# Chiral Edge States Emerging on Anyon-Net

Atsushi Ueda[1*], Kansei Inamura[2] Kantaro Ohmori[3]

**1** Department of Physics and Astronomy, University of Ghent, 9000 Ghent, Belgium
**2** Institute for Solid State Physics, University of Tokyo, Kashiwa 277-8581, Japan
**3** Faculty of Science, University of Tokyo, Tokyo 113-0033, Japan

⋆ Atsushi.Ueda@ugent.be

5th November 2024

## Abstract

We propose a symmetry-based approach to constructing lattice models for chiral topological phases, focusing on non-Abelian anyons. Using a 2+1D version of anyon chains and modular tensor categories(MTCs), we ensure exact MTC symmetry at the microscopic level. Numerical simulations using tensor networks demonstrate chiral edge modes for topological phases with Ising and Fibonacci anyons. Our method contrasts with conventional solvability approaches, providing a new theoretical avenue to explore strongly coupled 2+1D systems, revealing chiral edge states in non-Abelian anyonic systems.

# 1   Introduction

*Topologically ordered phases* in 2+1D systems—exemplified by fractional quantum Hall states [1, 2]—have motivated extensive research for decades. These phases are characterized by quasiparticles with distinctive statistics, called *anyons*, which are described by a mathematical framework known as a *modular tensor category* (MTC) [3]. A key objective in this field is to construct lattice models that can realize such topological phases. Levin and Wen provided a universal construction for non-chiral topological phases using commuting projector Hamiltonians [4], which generalize Kitaev's quantum double model [5]. Nevertheless, certain chiral topological phases, exhibiting gapless *chiral edge modes*, cannot be realized by commuting projector Hamiltonians on the lattice with a finite-dimensional Hilbert space on each site [3, 6–8]. As such, constructing lattice models that achieve general chiral topological phases continues to be a challenging and intriguing goal.

In this paper, we propose and study a *generalized symmetry* based approach to this problem. This approach is based on the perspective that the anyons can be understood as the spontaneous symmetry breaking of non-invertible one-form symmetry [9–11], described by the same MTC that consists of the anyon data. The idea of our construction is to realize the symmetry exactly at the microscopic lattice level. Such a microscopic MTC symmetry guarantees the topological anyon lines with desired statistics. While this approach is not new for Abelian anyons, as it is exploited in the pioneering work of Kitaev [3], generalization to non-Abelian anyons is recently proposed in [12], and in this paper, we perform numerical simulations to explicitly demonstrate the idea [1].

Specifically, we propose using a 2+1-dimensional version of anyon chains (see for example [13–16] for the 1+1D anyon chains or the corresponding 2+0D classical statistical models and [12] for their 2+1D/3+0D generalizations) to construct candidate models for non-Abelian chiral topological phases. This anyon model can be thought of as an anyonic analog of the chiral spin liquid model [17–26, 26, 27, 27–37], where spins are replaced by anyonic degrees of freedom. The model takes an MTC as input, and the dynamical variables on the lattice are based on the data of the MTC.

We perform numerical simulations of the model for the cases with the Ising category and the Fibonacci category as input MTCs. We use the matrix product state (MPS) [38–41] based density matrix renormalization group (DMRG) method [42] to simulate the system on an open square lattice. This is done by regarding the 2+1D model as a 1+1D model with long-range interactions and regarding anyon chains directly as an MPS [43].

For each of the MTCs, we found interesting phase structures depending on a parameter in the Hamiltonian we considered. In certain phases, we observed evidence for chiral edge modes: the entanglement entropy scaling consistent with the central charge of the chiral CFT, and the entanglement spectrum that matches the operator contents of the CFT. We also observed the chiral excitations are localized around the edge as shown in Fig. 1, providing further evidence of chiral behavior [2].

Conventionally, constructing chiral non-Abelian topological phases often requires exact or approximate solvability of the system. Examples include Kitaev's honeycomb model [3], the coupled wire construction [45] and their variants, string-net construction internal to chiral topological phases [46], wave functions based on chiral conformal field theories [47, 48], the

---

[1]The argument in [12] does not guarantee the chirality; the system can be in a non-chiral phase including the input MTC as a chiral half. Thus, the numerical simulations are necessary to confirm that the model can indeed realize a chiral topological phase.

[2]The local energy density is obtained by taking the expectation value for the local Hamiltonian term by the ground state and the excited states, and we define the excitation energy density $\delta E_{loc}$ as the difference thereof. Similarly, the local chirality is a phase of the expectation value of the local $\hat{h}_p$ operator defined by eq. (2). In this figure, we plot its difference for the ground state and the second excited state that is accessible through DMRG [44].

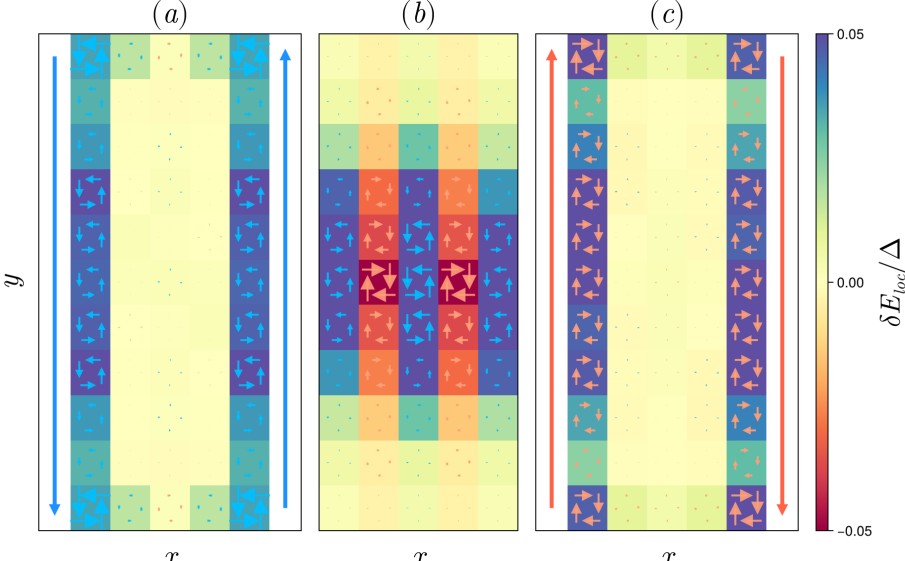

Figure 1: Real-space configuration of $(a)$ left-chiral, $(b)$ non-chiral, and $(c)$ right-chiral phases. The heatmaps represent the local excitation energy density $\delta E_{\text{loc}}$ normalized by the energy gap $\Delta$ for $\theta/\pi = (a)$ 0.1, $(b)$ 0.5, and $(c)$ 0.9 of the Fibonacci-type model. The excitation is localized on the edge for $\theta/\pi = 0.1$ and $\theta/\pi = 0.9$, while it spreads into the bulk for $\theta/\pi = 0.5$. The arrows represent the local chirality relative to the ground state, with their color representing the chirality and their size scaling according to the amplitude.

surface of 3+1D invertible states [49–52], and others. In contrast, our approach is founded on the exact symmetry of the microscopic lattice model, functioning within a strongly coupled, isotropic, and microscopically homogeneous regime. Furthermore, our method, in principle, applies to any MTC up to the limits imposed by numerical computation. Thus, we consider our work a new theoretical avenue that complements existing studies by offering a method to microscopically study topological phases.

## 2   Model.

We consider a model of interacting anyons on a square lattice with the open boundary condition. The number of lattice sites in the $x$ and $y$ directions are denoted by $W$ and $H$, respectively. The input datum of the model is a pair $(\mathcal{B}, \rho)$ of a modular tensor category $\mathcal{B}$ and an arbitrary object $\rho \in \mathcal{B}$ [3]. Given a pair $(\mathcal{B}, \rho)$, one can define the state space of the model and write down the Hamiltonian in terms of anyon diagrams of $\mathcal{B}$.

To define the state space, we consider the anyon diagrams on a square lattice as depicted in Fig. 2. The edges and vertices of the lattice are labeled by simple objects and basis morphisms that are compatible with the fusion rules of $\mathcal{B}$. These objects and morphisms are regarded as dynamical variables of the model. The state space $\mathcal{H}$ is spanned by all possible configurations of these dynamical variables. We note that $\mathcal{H}$ is not the tensor product of local state spaces due to the constraints from the fusion rules. Nevertheless, since the constraints are local, one can impose these constraints energetically and realize $\mathcal{H}$ as a low-energy sector of a tensor product state space.

---

[3]The model can also be defined for general braided fusion category $\mathcal{B}$. This paper focuses on the case where $\mathcal{B}$ is modular.

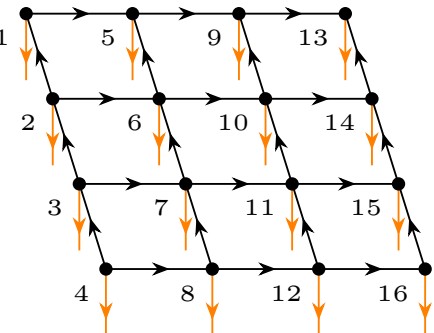

Figure 2: The above anyon diagram represents a state on a square lattice with $W = H = 4$. Here, the vertical lines are labeled by the chosen object $\rho$.

The Hamiltonian of the model is given by

$$H = \sum_{p:\text{plaquetts}} \left( e^{i\theta} \hat{h}_p + e^{-i\theta} \hat{h}_p^\dagger \right) - g \sum_{p:\text{plaquetts}} \hat{B}_p, \tag{1}$$

where $\theta$ and $g$ are positive real numbers. Here, $\hat{B}_p$ is the Levin-Wen plaquette operator defined by [4]

$$\hat{B}_p \quad \cdots \quad = \sum_{a \in \text{Simp}(\mathcal{B})} \frac{d_a}{\sqrt{\mathcal{D}}} \quad \cdots \quad ,$$

where $d_a$ is the quantum dimension of $a \in \mathcal{B}$ and $\mathcal{D} = \sum_{a \in \text{Simp}(\mathcal{B})} d_a^2$ is the total dimension of $\mathcal{B}$. The summation on the right-hand side is taken over all simple objects of $\mathcal{B}$. The first term $\hat{h}_p$ of the Hamiltonian (1) is defined by the 90° rotation of the plaquette $p$ as follows:

$$\hat{h}_p \quad \cdots \quad = \quad \cdots \quad . \tag{2}$$

Its Hermitian conjugate $\hat{h}_p^\dagger$ is given by the 90° rotation in the opposite direction. We emphasize that the Hamiltonian (1) generically breaks the time-reversal symmetry due to the non-trivial braiding statistics of anyons in $\mathcal{B}$ [4].

Precisely, the operator $\hat{h}_p : \mathcal{H} \to \mathcal{H}$ defined by Eq. (2) is ambiguous because there are many different ways to evaluate the diagram on the right-hand side of Eq. (2). To eliminate this ambiguity, one needs to fix a convention for evaluating the right-hand side of Eq. (2). The resulting model no longer involves any ambiguities. However, the model now depends on the convention one chooses.

---

[4]We can show this by contradiction. To see this, let us suppose that the Hamiltonian has a time-reversal symmetry $T$. Since $T$ is anti-unitary, the invariance of $H$ under $T$ implies that there exists a unitary operator $U$ such that $U\hat{h}_p U^\dagger = \hat{h}_p^\dagger$. This, in turn, implies that $\hat{h}_p$ and $\hat{h}_p^\dagger$ should have the same set of eigenvalues. However, this does not hold for a generic MTC. Thus, the Hamiltonian (1) does not have a time-reversal symmetry in general.

The model becomes independent of a convention in the limit where $g \to \infty$. In this limit, the state space $\mathcal{H}$ effectively reduces to a subspace $\mathcal{H}_0 \subset \mathcal{H}$ on which $\hat{B}_p = 1$ for all plaquettes [5]. The reduced state space $\mathcal{H}_0$ is isomorphic to the Hom space

$$\mathcal{H}_0 \cong \mathrm{Hom}_{\mathcal{B}}(\mathbb{1}, \rho^{\otimes WH}), \tag{3}$$

where $\mathbb{1}$ denotes the unit object (i.e., the trivial anyon) of $\mathcal{B}$. On this subspace, the operator $\hat{h}_p$ does not depend on how to evaluate the diagram in Eq. (2) due to the pentagon and hexagon equations of $\mathcal{B}$. In what follows, we focus on the model with $g = \infty$ and identify the state space $\mathcal{H}_0$ with $\mathrm{Hom}_{\mathcal{B}}(\mathbb{1}, \rho^{\otimes WH})$.

For numerical calculations, it is convenient to regard our model as a 1+1D model with long-range interactions. Specifically, a state $|\psi\rangle \in \mathcal{H}_0$ is represented by a one-dimensional array of $\rho$-anyons that fuse into a trivial anyon $\mathbb{1}$ as follows:

$$|\psi\rangle = \qquad \text{.} \tag{4}$$

Here, we draw the case where $W = H = 4$ for simplicity. The labeles $\{1, 2, 3, \cdots, WH = 16\}$ specify which anyons in Eq. (4) correspond to which anyons on the square lattice in Fig. 2. The Hamiltonian (1) acting on the state (4) is represented by the braiding of anyons. For example, the operator $\hat{h}_p$ for a plaquette $p = (1256)$ of a $4 \times 4$ square lattice is given by

$$\hat{h}_p |\psi\rangle = \qquad \text{.} \tag{5}$$

We note that anyons 3 and 4 are in front of anyons 1, 2, 5, and 6 as shown in Fig. 2.

A remarkable feature of our model is the exact non-invertible 1-form symmetry described by the input MTC $\mathcal{B}$ [12]. This symmetry is always spontaneously broken due to the non-degeneracy of the modular $S$-matrix [6]. Accordingly, if gapped, the ground state of our model must exhibit a topological order that contains a subsector described by $\mathcal{B}$. That is, our model realizes a topological order $\mathcal{B} \boxtimes \mathcal{C}$, where $\mathcal{C}$ is another MTC. This follows from the fact that an MTC $\mathcal{B}'$ that contains another MTC $\mathcal{B}$ as a subcategory is braided-equivalent to the product of two MTCs $\mathcal{B}$ and its centralizer $\mathcal{C}$ in $\mathcal{B}'$ [53].

Our model has a natural interpretation as a system of interacting anyons in a parent topological order [7]. The parent topological order is realized by the Levin-Wen model with the input being an MTC $\mathcal{B}$. Indeed, the state space $\mathcal{H}_0$ given by Eq. (3) is the ground state subspace of the Levin-Wen model on a disk in the presence of a bunch of $\rho$-anyons. The Hamiltonian (1) describes the interactions of these anyons, which can drive the system into other topologically ordered phases.

The anyons of the parent topological order are described by the Drinfeld center $Z(\mathcal{B}) = \mathcal{B} \boxtimes \overline{\mathcal{B}}$, where $\overline{\mathcal{B}}$ denotes the time-reversal of $\mathcal{B}$. Since the topological order $\mathcal{B} \subset Z(\mathcal{B})$ is guaranteed

---

[5]The Levin-Wen plaquette operators $\hat{B}_p$ are commuting projectors and hence admit a simultaneous eigenstate with eigenvalues 1.

[6]A 1-form symmetry is said to be spontaneously broken if there exists a topological line operator that is charged under the symmetry operator. The non-degeneracy of the modular $S$-matrix implies that for each symmetry operator, there exists a topological line charged under it.

[7]The idea of constructing models of interacting anyons in a parent topological phase was employed in the original anyon chain model [13].

by the symmetry of the model, it cannot be violated by interactions. On the other hand, the topological order $\overline{\mathcal{B}} \subset Z(\mathcal{B})$ is emergent and hence can be affected by the interactions. When the interactions turn $\overline{\mathcal{B}}$ into another topological order $\mathcal{C}$, the model realizes the topological order $\mathcal{B} \boxtimes \mathcal{C}$. In particular, the model exhibits a chiral topological order if $\mathcal{C}$ and $\overline{\mathcal{B}}$ have different chiral central charges.

In the subsequent section, we will exclusively work on the cases where $\mathcal{B}$ is either the Fibonacci category or the Ising category. The Fibonacci category consists of two simple objects $\{\mathbb{1}, \tau\}$ that obey the fusion rule

$$\tau^2 = \mathbb{1} \oplus \tau. \tag{6}$$

On the other hand, the Ising category consists of three simple objects $\{\mathbb{1}, \eta, \sigma\}$ that obey the fusion rules

$$\eta^2 = \mathbb{1}, \quad \eta \otimes \sigma = \sigma \otimes \eta = \sigma, \quad \sigma^2 = \mathbb{1} \oplus \eta. \tag{7}$$

## 3   Tensor network simulations.

We employ tensor networks to investigate the physical properties delineated by the Hamiltonian (1). Our goal is to variationally optimize the ground state within this framework. We initiate this process by reconfiguring the fusion tree described in Eq. (4), incorporating variational degrees of freedom as detailed in Ref. [43]. Specifically, we expand the state (4) by selecting basis sets conditioned on the choice of fusion channel $e_n$, depicted in the following diagram [8]:

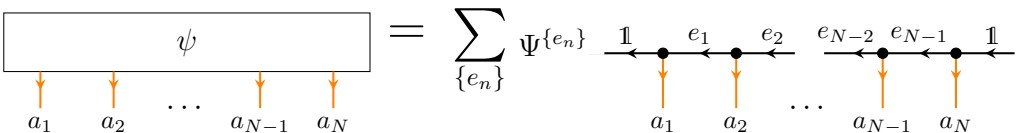

In this formulation, the coefficients $\Psi^{\{e_n\}}$ are defined based on the selected fusion channels $\{e_n\}$, which act as dynamical variables. This high-rank tensor $\Psi^{\{e_n\}}$ is subsequently factorized into a product of matrices:

$$\Psi^{\{e_n\}} = A^{\mathbb{1}e_1}_{1\mu_1} A^{e_1 e_2}_{\mu_1 \mu_2} A^{e_2 e_3}_{\mu_2 \mu_3} \cdots A^{e_{N-2}e_{N-1}}_{\mu_{N-2}\mu_{N-1}} A^{e_{N-1}\mathbb{1}}_{\mu_{N-1}1},$$

where each factor $A^{e_{n-1}e_n}$ is an $N^{(e_{n-1})}_{n-1} \times N^{(e_n)}_n$ matrix, i.e., the subscript $\mu_n$ runs from 1 to $N^{(e_n)}_n \in \mathbb{Z}$. Throughout our study, all fusion labels $a_n$ are designated as $\rho$, enabling us to employ matrix representations as illustrated below:

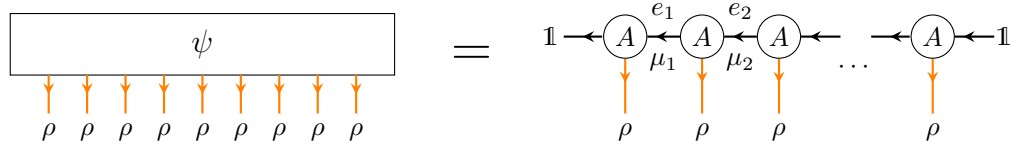

For example, considering Fibonacci anyons where $\rho = \tau$, the selection rule necessitates $e_1 = \tau$ since $\mathbb{1} \otimes \tau = \tau$. Similarly, $e_2$, $e_3$, and $e_4$ can take values $\mathbb{1}$ or $\tau$, following the fusion rules $\tau^2 = \mathbb{1} \oplus \tau$, $\tau^3 = \mathbb{1} \oplus 2\tau$, and $\tau^4 = 2\mathbb{1} \oplus 3\tau$. The coefficients of the simple objects correspond to the range of $\mu_n$. Thus, for precise decomposition, $N^{(\mathbb{1})}_4$ and $N^{(\tau)}_4$ must be 2 and 3, respectively. As the total dimension $N^{(\mathbb{1})}_n + N^{(\tau)}_n$ scales exponentially with system size, we impose a cutoff

---

[8]Technically, here we suppose that the MTC $\mathcal{B}$ is multiplicity free, i.e., all fusion coefficients are 0 or 1. This condition is satisfied in both the Fibonacci and Ising categories.

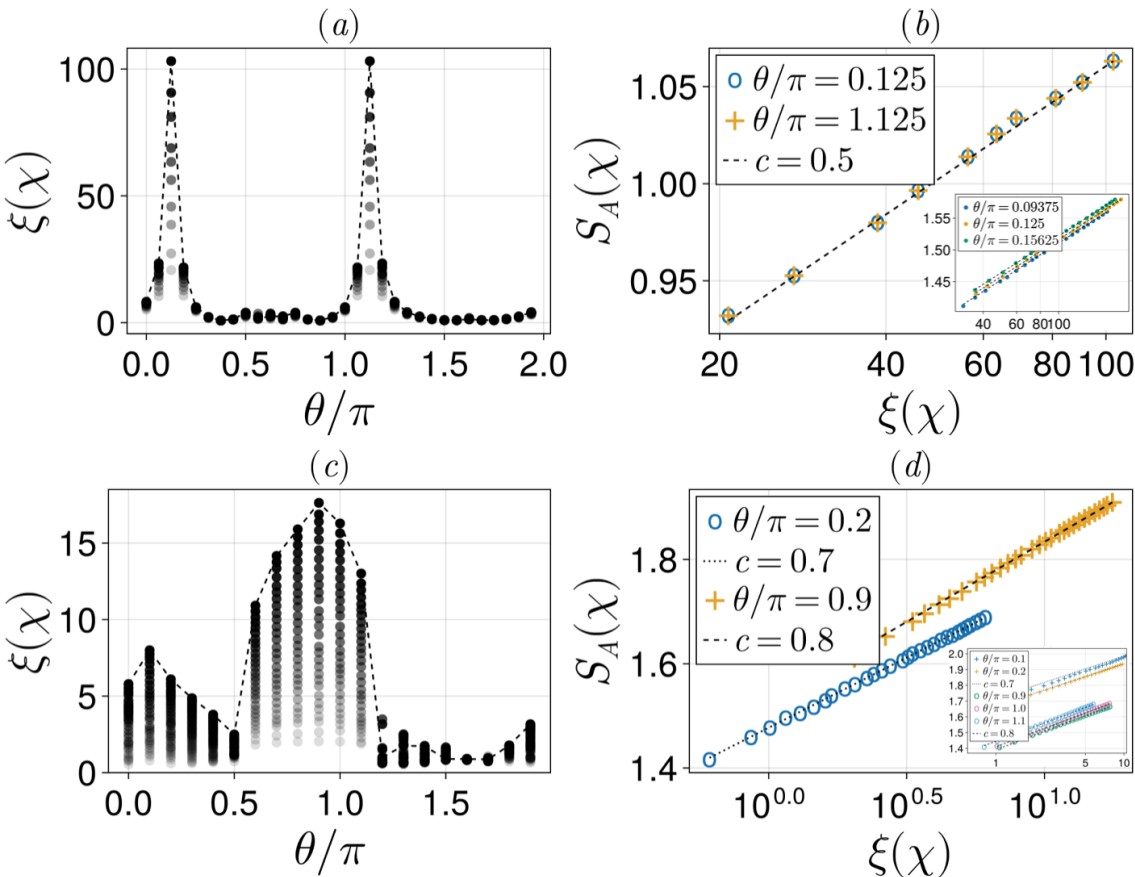

Figure 3: The correlation length of the ground states for $(a)$ the Ising-type($H = 4$) and $(c)$ Fibonacci-type($H = 5$) models. The correlation length was calculated from the second/third leading eigenvalues of the MPS transfer matrix with $\chi = 24 \sim 60/24 \sim 140$, respectively. The data points from larger $\chi$ are indicated with darker black. The entanglement entropy of the half strip grows as $S_A \sim \frac{c}{6} \ln(\xi)$ in the gapless regions of $(b)$ the Ising-type and $(d)$ Fibonacci-type model. The inserts show $H = 6$ results with larger $\chi \sim 240$, indicating the gapless regions extend as phases.

total bond dimension $\chi$ to ensure computational feasibility. For the Ising case, the total bond dimensions correspond to $N_n^{(\mathbb{1})} + N_n^{(\eta)}$ and $N_n^{(\sigma)}$ [9]. In our numerical simulations, $\chi$ is equally distributed for each simple object. These low-rank representations of the states, commonly known as MPS, enable us to variationally optimize these matrices. This approach facilitates the accurate determination of ground states while maintaining numerical traceability.

In our MPS simulations, we consider a system where the width, denoted as $W$, is much larger than the height $H$. In this limit, the dominant edge modes, if present, will likely be localized on the upper and lower edges of the strip, allowing us to treat the system as quasi-one-dimensional for efficient simulation using tensor network methods. However, finite-size DMRG simulations often yield unclear results due to the influence of edge modes on the left and right sides of the strip. To address this, we employ variational uniform matrix product states (VUMPS) as described in [54], which effectively consider $W \to \infty$ and thereby sidestep these edge effects to focus on the upper and lower modes. Nevertheless, the system is not truly infinite but rather finitely correlated: the bond dimension $\chi$, which controls accuracy, imposes a finite correlation length $\xi(\chi)$. As $\chi$ is sufficiently increased, $\xi(\chi)$ approaches the true correlation length of the

---

[9]In the case of the Ising category, we choose $\rho = \sigma$. For this choice, the fusion rules (7) force $e_n = \mathbb{1}, \eta$ for odd $n$ and $e_n = \sigma$ for even $n$. In particular, the model is well-defined only when the number of $\rho$'s is even.

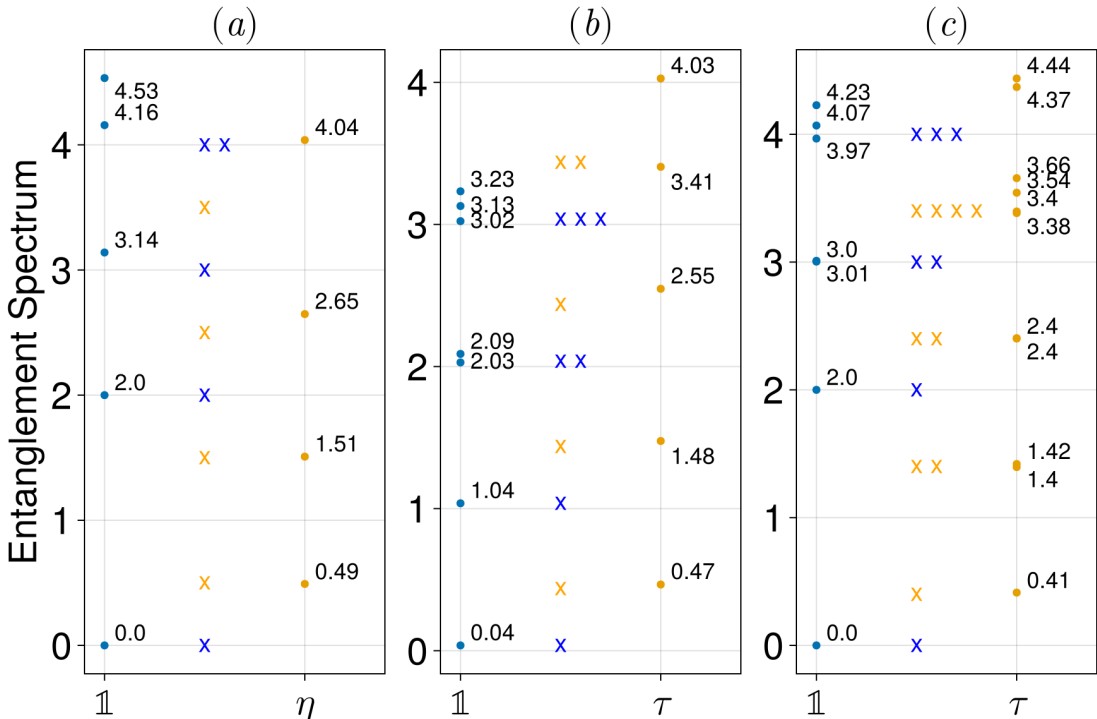

Figure 4: Entanglement spectrum of ($a$) the Ising-type with $\theta = \frac{\pi}{8}(H = 4)$ and the Fibonacci-type anyon model at ($b$) $\theta = 0.2\pi(H = 6)$ and ($c$) $\theta = 0.9\pi(H = 6)$. All spectra are normalized so that the first excitations in the $\mathbb{1}$ sector match the theoretical prediction. The theoretical boundary CFT spectrum and multiplicity are presented with crosses.

original model. The exception occurs in gapless systems where the correlation length diverges. In this case, $\xi(\chi)$ continuously increase with $\chi$ . Nevertheless, the scaling of the physical properties with $\xi(\chi)$ can be used to detect universal properties, which is often referred to as finite-entanglement scaling [55–61]. To elaborate, if the system exhibits a gapless spectrum described by conformal field theory (CFT), the entanglement entropy scales as

$$S_A(\chi) \sim \frac{c}{6} \ln \xi(\chi), \tag{8}$$

enabling the detection of the central charge [62].

In this study, we consider two models: the Ising-type with $\rho = \sigma$ and the Fibonacci-type with $\rho = \tau$. Figure 3 presents the numerical results obtained from VUMPS. For most of the parameter space, the correlation length $\xi(\chi)$ converges as $\chi$ increases, indicating finite energy gaps in the thermodynamic limit. However, the correlation length continues to grow at $\theta = \frac{\pi}{8}, \frac{9\pi}{8}$ for the Ising-type, and around $\theta = 0.1\pi, 0.9\pi$ for the Fibonacci-type models, respectively. In these regions, the scaling described in Eq. (8) is observed, which allows us to determine the central charges $c = 0.5, 0.7$, and $0.8$. These gapless regions expand as the strip width is increased, as shown in the insets.

To further characterize the gapless regions, we analyze the entanglement spectrum. Under conformal invariance, the entanglement spectrum is proportional to the boundary scaling dimension, allowing for the deduction of the operator contents of the theory [63–66].

We find that the operator contents observed in our calculations align with those of the free-free boundary conditions of the Ising CFT [67], the Ramond sector of the (superconformal) tricritical Ising CFT [68, 69], and the AB-AB boundary conditions of the three-state Potts

CFT [70], respectively (Fig. 4). The partition functions of these CFTs are given by

$$Z_{\text{free,free}}^{\text{Ising}} = \chi_I + \chi_\epsilon,$$

$$Z_{\text{Ramond}}^{\text{tricritical}} = \chi_\sigma + \chi_{\sigma'}, \quad Z_{\text{AB,AB}}^{\text{3-state Potts}} = \chi_I + \chi_\epsilon, \tag{9}$$

where $\chi_I = \chi_{1,1}^{c=1/2}$ and $\chi_\epsilon = \chi_{2,1}^{c=1/2}$ for the Ising CFT, $\chi_\sigma = \chi_{1,2}^{c=7/10}$ and $\chi_{\sigma'} = \chi_{2,1}^{c=7/10}$ for the tricritical Ising CFT, and for the three-state Potts model, $\chi_I = \chi_{1,1}^{c=4/5} + \chi_{4,1}^{c=4/5}$ and $\chi_\epsilon = \chi_{2,1}^{c=4/5} + \chi_{3,1}^{c=4/5}$. Here, $\chi_{r,s}^c$ denotes the character of representation $(r,s)$ of the Virasoro algebra with central charge $c$. Using the small-$q$ expansions of the characters (see the Appendix), the low-lying spectra of the boundary scaling dimensions for the Ising, the tricritical Ising, and three-state Potts CFT are respectively $(0, 0.5, 1.5, 2, ...)$, $(\frac{3}{80}, \frac{7}{16}, \frac{83}{80}, \frac{23}{16}, ...)$, and $(0, 0.4, 1.4, 1.4, 2, ...)$. On the other hand, we obtain the entanglement spectrum from a Schmidt decomposition of the ground state as $|\psi_{GS}\rangle = \sum_n \Lambda_n |\psi_{GS}^{(n)}\rangle_L \otimes |\psi_{GS}^{(n)}\rangle_R$, where the entanglement spectrum is defined as $-\ln \Lambda_n$ up to normalization. ($\Lambda_n$ has a label by the simple object in the channel of decomposition. See Ref. [43] for a detailed treatment of bipartite decompositions of anyonic systems.) The numerical entanglement spectrum, illustrated in Fig. 4, matches well with the CFT predictions (denoted with crosses), confirming that the gapless edge states are described by these CFTs.

Further evidence of the chiral edge states through exact diagonalization, in addition to DMRG (Fig. 1), is provided in the appendix.

The chiral edge modes indicated by the above numerical results suggest that our lattice model realizes chiral topological phases. Specifically, for the Fibonacci-type model at $\theta = 0.2\pi$, our result is consistent with the bulk topological order Fib $\boxtimes \overline{\text{Spin}(7)_1}$. Here, Fib denotes the Fibonacci category with chiral central charge $c_- = 2.8$, while $\text{Spin}(7)_1$ denotes the MTC that consists of three anyons $\{\mathbb{1}, \eta, \sigma\}$ obeying the Ising fusion rules (7) and has chiral central charge $c_- = 3.5$ [10]. Similarly, for the Fibonacci-type model at $\theta = 0.9\pi$, our result is consistent with the bulk topological order Fib $\boxtimes \overline{\text{SU}(3)_1}$, where $\text{SU}(3)_1$ is the $\mathbb{Z}_3$ abelian topological order with chiral central charge $c_- = 2$. In both cases, the existence of the Fibonacci anyon is guaranteed by the exact 1-form symmetry of the model. Since the chiral central charge is additive, Fib $\boxtimes \overline{\text{Spin}(7)_1}$ and Fib $\boxtimes \overline{\text{SU}(3)_1}$ have chiral central charges $-0.7$ and $0.8$, respectively. The opposite signs of these values imply that the associated gapless edge modes have the opposite chirality. This is consistent with the numerical result shown in Figs. 1 and the appendix. We note that similar chiral topological phases with the Fibonacci anyon have been investigated in the literature using the couple-wire-like construction [71–78].

On the other hand, for the Ising-type model, the numerical results in Figs. 3 and 4 suggest that the bulk exhibits the chiral Ising topological order or its time-reversal at $\theta = 0.125\pi$ and $\theta = 1.125\pi$. Naively, the time-reversal of the chiral Ising topological order seems to contradict the exact 1-form symmetry of the model because the input MTC is the Ising category rather than its time-reversal. Nevertheless, the exact diagonalization shown in the appendix indicates that the edge modes around $\theta = 0.125\pi$ and $\theta = 1.125\pi$ have the opposite chirality, implying that both Ising and $\overline{\text{Ising}}$ are realized in our model. We leave this puzzle for the future.

## 4   Conclusion and outlook.

In this paper, we have proposed a 2+1D lattice model of interacting anyons that realizes a chiral topological phase. The model is defined as a generalization of an anyon chain, and admits an exact non-invertible 1-form symmetry described by the input MTC at the microscopic level. We

---

[10]The chiral central charge of an MTC is defined modulo 8.

have performed numerical simulations of the model for the Ising and Fibonacci categories, and observed evidence for chiral edge modes in the ground states. This study shows that we can use the MTC symmetry, coupled with the numerical techniques, to explore the strongly coupled 2+1D systems without relying on exact or approximate solvability.

Our results pave the way for new theoretical studies of chiral phases in 2+1D systems. The model can incorporate any MTC as input, enabling the exploration of a wide range of topological phases. Another interesting direction is to examine the bulk properties in more detail. Our numerical simulations were conducted on open lattices. While this configuration reveals the chiral edge modes, it also obscures the bulk quantities. Studying systems with periodic boundary conditions would provide clearer insights into the bulk phase. Lastly, numerical simulation of higher dimensional anyonic system can have applications beyond topological phases. For example, [79, 80] proposed to use anyon-based model to simulate Yang-Mills theory on lattice, and our numerical methods in principle would apply to the high-energy/hadron physics context.

## 5   Acknowledgements

We thank Linhao Li, Kevin Vervoort for stimulating discussions. In particular, A.U. would like to thank Lukas Devos for the help with the MPSKit [81]. We are convinced that MPSKit will be a natural framework for investigating strongly coupled anyonic systems.

**Funding information**   A.U. was supported in part by Watanabe Foundation. K.I. was partly supported by FoPM, WINGS Program, the University of Tokyo, and also by JSPS Research Fellowship for Young Scientists. KO is supported by JSPS KAKENHI Grant-in-Aid No.22K13969 and No.24K00522. K.O. also acknowledges support from the Simons Foundation Grant #888984 (Simons Collaboration on Global Categorical Symmetries). A part of the computation in this paper has been done using the facilities of the Supercomputer Center, the Institute for Solid State Physics, the University of Tokyo.

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

## .1   Exact diagonalization

Accessing the higher excitation spectrum is limited in DMRG. Consequently, we have exactly diagonalized the $4 \times 4$ Hamiltonian to further support the existence of chiral edge states. To characterize the chirality, we define the momentum of the edge state as follows. In cases of translational invariant states in 1D, a state acquires an $e^{ik}$ factor after a one-site translation $\hat{T}$. If present, the edge states can be conceptualized as forming a one-dimensional ring, thereby allowing us to define the one-site translation of the edge, denoted as $\hat{T}_{\text{edge}}$, to obtain the momentum as the phase of the expectation value. In this study, we define $\hat{T}_{\text{edge}}$ as the braiding of twelve $\rho$s on the edges in a counter-clockwise direction. Figure. 5 demonstrates the numerically obtained dispersion for the Ising and Fibonacci type Hamiltonian. The results in the main text shown in Fig. 3 suggest the existence of gapless chiral edge states around $\theta = 0.125\pi, 1.125\pi$ and $\theta = 0.2\pi, 0.9\pi$ for the Ising and Fibonacci cases, respectively. The energy dispersion in Fig. 5 is consistent with this picture. While the small system size does not allow for sharp determination of the phase boundaries, the boundaries from VUMPS are consistent with the exact diagonalization.

## .2   Small-$q$ expansions of the characters

In CFT, the partition function on a torus is a function of the modular parameter $\tau$ and $q = e^{2\pi i \tau}$. From the exponents of $q$, we can read off the scaling dimensions of operators of the theory. The small-$q$ expansions of the characters in eq. (9) are [82]:

$$
\begin{aligned}
Z_{\text{free,free}}^{\text{Ising}} &= q^{-\frac{c}{24}} \left( 1 + q^2 + q^3 + 2q^4 + \cdots \right) \\
&\quad + q^{-\frac{c}{24}+\frac{1}{2}} \left( 1 + q + q^2 + q^3 + 2q^4 + \cdots \right), \\
Z_{\text{Ramond}}^{\text{tricritical}} &= q^{-\frac{c}{24}+\frac{3}{80}} \left( 1 + q + 2q^2 + 3q^3 + \cdots \right) \\
&\quad + q^{-\frac{c}{24}+\frac{7}{16}} \left( 1 + q + q^2 + 2q^3 + \cdots \right), \\
Z_{\text{AB,AB}}^{\text{3-state Potts}} &= q^{-\frac{c}{24}} \left( 1 + q^2 + 2q^3 + 3q^4 + \cdots \right) \\
&\quad + q^{-\frac{c}{24}+\frac{2}{5}} \left( 1 + 2q + 2q^2 + 4q^3 + \cdots \right).
\end{aligned}
$$

The boundary scaling dimensions correspond to the relative exponents of $q$ to the ground state if the CFT is unitary. Thus, the scaling dimension and the degeneracy can be read off from the exponents of $q$ (after taking out $q^{-\frac{c}{24}}$) and their prefactors. In fact, these exponents are consistent with those of the corresponding primary operators as below.

| Ising | | Tricritical Ising | | 3-state Potts | |
|---|---|---|---|---|---|
| $\Delta_I$ | $0$ | $\Delta_\sigma$ | $\frac{3}{80}$ | $\Delta_I$ | $0$ |
| $\Delta_\epsilon$ | $\frac{1}{2}$ | $\Delta_{\sigma'}$ | $\frac{7}{16}$ | $\Delta_\epsilon$ | $\frac{2}{5}$ |

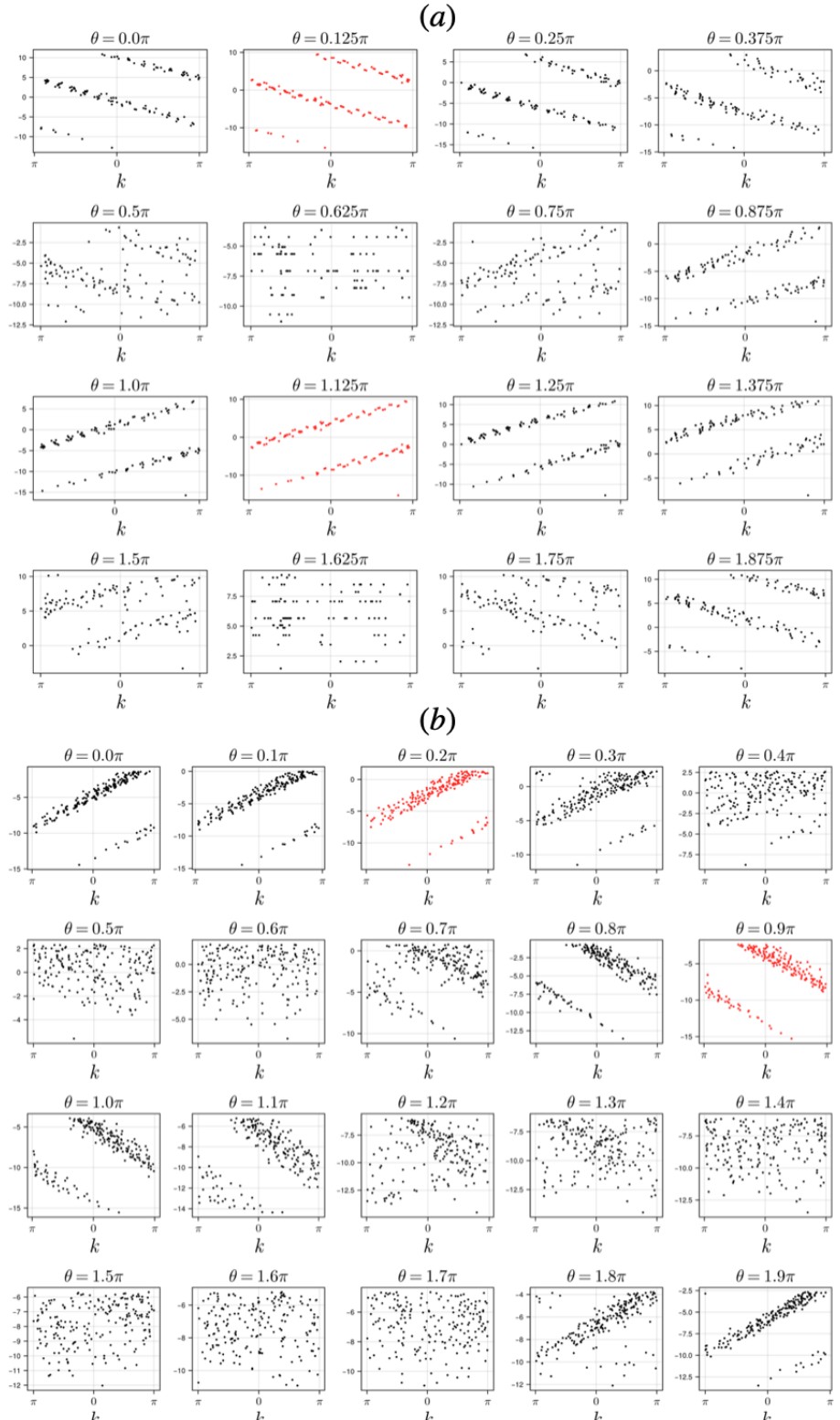

Figure 5: The energy spectrum of the 4×4 (*a*) Ising-type and (*b*) Fibonacci-type models obtained by exact diagonalization. The horizontal axis $k$ is the "quasi-momentum" of the edge states. Chiral energy-momentum dispersion is observed in the same region as discussed in the main text (denoted with red points).