# Peer review of "Chiral Edge States Emerging on Anyon-Net"

_SciPost Physics Core_

## Round 2 · Referee Report · Anonymous (Referee 1) · 2024-11-13

Strengths

1- new construction of lattice model that stabilizes non-abelian chiral topological order 2- strong numerics with state-of-the-art tensor network methods 3- convincing signatures of topological order

Weaknesses

1- determination of the phase diagram is not established

Report

In this work, the authors introduce lattice models that realize phases with non-abelian chiral topological order. The idea for the construction, which is (2+1)D generalization of anyon chains, was introduced in an earlier work by some of the authors, but there the occurence of chiral topological phases was only hinted at. In this work, therefore, the authors consider a model with an explicitly chiral term, and use numerical simulations to show that the model exhibits chiral topological order for parts of the phase diagram.

The exploration of lattice models with (non-abelian) chiral topological order is an active research direction, in the context of spin liquids and fractional Chern insulators. In that respect, this work opens up a new, important, avenue for further explorations. In particular, the MTC symmetry is introduced explicitly at the microscopic level, which makes the construction very transparent and interesting.

The numerical simulations make use of tensor network parametrizations that explicitly encode the anyonic symmetry of the microscopic model, which is definitely the state-of-the-art method for simulating these types of models. The simulations are convincing that there is a part of the phase diagram that stabilizes a chiral topological order, as showcased beautifully by the extraction of the central charge and the entanglement spectrum on the open strip geometry.

For these reasons, I recommend publication of this manuscript.

I have one concern with the author's claim that there is a region in the phase diagram where the topological phase is stabilized, based on the results in Fig. 3. At the smallest values of H (in panels a and c), the divergence of the correlation length seems to be at a single point. In the inset of panels b and d, there is evidence that the central charge scaling is observed for more points in the vicinity as well. I don't think this is clear evidence for an extended phase: an equally plausible explanation is that these different points are all in the scaling region of this single critical point. Based on the numerical evidence, therefore, I don't believe the authors can make any definite claim about the topological phase boundaries.

I was wondering why the authors did not consider the use of cylindrical boundary conditions instead of the strip geometry. In that set-up, the phase boundaries and phase transitions should be more clearly visible and there's other signatures of topological order such as k-resolved entanglement spectra and topological entanglement entropy. This is rather standard practice in the numerical study of chiral spin liquids and fractional Chern insulators.

Requested changes

1- Reconsider claims on the phase diagrams for both models

Recommendation

Publish (easily meets expectations and criteria for this Journal; among top 50%)

  • validity: high
  • significance: high
  • originality: high
  • clarity: top
  • formatting: perfect
  • grammar: perfect

Author:  Atsushi Ueda  on 2025-05-13  [id 5475]

(in reply to Report 1 on 2024-11-13)

We appreciate your comments. Here are the list of our replies.

I have one concern with the author's claim that there is a region in the phase diagram where the topological phase is stabilized, based on the results in Fig. 3. At the smallest values of H (in panels a and c), the divergence of the correlation length seems to be at a single point. In the inset of panels b and d, there is evidence that the central charge scaling is observed for more points in the vicinity as well. I don't think this is clear evidence for an extended phase: an equally plausible explanation is that these different points are all in the scaling region of this single critical point. Based on the numerical evidence, therefore, I don't believe the authors can make any definite claim about the topological phase boundaries.

We appreciate your concerns. We found that it is a fair point. The scenario with point-like topological phase is certainly possible from the current numerical data. The corresponding sentences are modified accordingly.

I was wondering why the authors did not consider the use of cylindrical boundary conditions instead of the strip geometry. In that set-up, the phase boundaries and phase transitions should be more clearly visible and there's other signatures of topological order such as k-resolved entanglement spectra and topological entanglement entropy. This is rather standard practice in the numerical study of chiral spin liquids and fractional Chern insulators.

In conventional spin systems, the notion of periodic boundary conditions is well-established. On the other hand, anyonic chains and their generalizations involve subtlety in adding periodic terms as it braids through the anyons. The complication increases more in two dimensions. It is also simply numerically challenging because the interactions between boundaries are converted to long-range interactions in DMRG. We avoided it by using open boundaries. While the bulk properties are less clear, the edge states can be used as a detector of topological phases.

---

## Round 2 · Referee Report · Juraj Hasik (Referee 2) · 2025-2-4

Strengths

  • novel bottoms-up approach to prepare topological phases with chiral edge modes
  • the strategy introduced here, with anyons being the elementary objects, can be used to explore chiral topological phases even if the underlying microscopic model is not (yet) known

Weaknesses

  • Accessibility can be improved by additional explanation of some categorical terms/concepts [Hom space, Drinfeld center, $\boxtimes$ symbol]
  • construction breaks time-reversal symmetry, thus unable to address scenarios with spotaneous chiral topological order

Report

The manuscript introduces a 2D model, formulated entirely in the languages of anyons as elementary objects, capable of realizing phases with chiral edges. A numerical approach, based on tensor networks, is then used to study two representative examples based on Fibonnaci and Ising MTCs, allowing for explicit enforcement of fusion rules. The authors present a strategy on how to carry out the analysis. First, a scaling of correlation length (with bond dimension) is used to identify gapped and gapless phases (presumably due to edges modes). Afterwards, using entanglement scaling and entanglement spectra, the CFT on the edge can be further identified - in this manuscript, together three instances are identified and discussed in further detail.

I believe this work presents a neat new approach to numerically simulate and study a wide zoo of chiral topological orders, importantly including non-abelian cases. Therefore I am happy to recommend it for publication.

Below are few clarifying questions/remarks

  • The $B_p=1$ is not enforced energetically here (through optimization) but rather by the allowed set of tensors A (labeled by $e_n$) that respect the fusion rules, right ?

  • How are the finite strips of Fig.1 mapped into 1D chains ? From the data, I would guess along the long dimension, width W, of the strip rather then height H, making the interactions longer ranged. Is that so ?

  • Entanglement spectra are calculated from infinite-W strips or cylinders ?

  • Values of bond dimensions used are relatively small compared to state-of-the-art in VUMPS/iDMRG simulations of spin models. Is this due to the overhead in handling anyons ? What is the leading comp. complexity here - would, say, H=10 case be computationally too prohibitive ?

  • footnote 9 states " ... force $e_n = 1,\eta$ for odd n and $e_n =\sigma$ for even n". Isn't it the other way around, i.e. for $e_2$ the $1\otimes\sigma\otimes\sigma = 1\oplus \eta$, so $e_n=1,\eta$ for even n ?

  • When discussing bulk phase, i.e. Fib⊠Spin$(7)_1$ for $\theta/\pi=0.2$ etc., these identifications are based on the entanglement spectra data (+ central charge) ? Or specifically, what (presumably emergent) structure in tensors A reveals this {1,η,σ} MTC ?

  • Here, every site is taken to host a non-trivial object. Should the $\Psi$ be seen as describing a number of WH anyons $\rho$ , i.e. a "full filling" ? Can one introduce a chemical potential for $\rho$-anyons ?

Requested changes

1- Could the authors expand upon the ambiguity of the evaluation of $h_p$ ? Is this due to various options in ordering of $\rho$ edges ?

2 - For $\theta/\pi=0.2$ region in the inset of Fig. 3 (c) the fit for $\theta/\pi=0.1$ slightly deviates from the data. Maybe adding $\theta/\pi$= 0.3 and 0.4 for H=5 one can appreciate the distinction between situations where scaling formula (8) applies with high precision and cases where the edges remain gapped. [Here caption says inserts, main text insets]

Recommendation

Publish (easily meets expectations and criteria for this Journal; among top 50%)

  • validity: top
  • significance: high
  • originality: high
  • clarity: good
  • formatting: excellent
  • grammar: excellent

Author:  Atsushi Ueda  on 2025-05-13  [id 5474]

(in reply to Report 2 by Juraj Hasik on 2025-02-04)

Thank you very much for the insightful comments. Here are the list of our answers.

The $B_p=1$is not enforced energetically here (through optimization) but rather by the allowed set of tensors A (labeled by $e_n$) that respect the fusion rules, right ?

Yes, it is correct.

How are the finite strips of Fig.1 mapped into 1D chains ? From the data, I would guess along the long dimension, width W, of the strip rather then height H, making the interactions longer ranged. Is that so ?

Yes, we ordered $WH$ points along the width of the rectangle. The details are now added in the footnote 11.

Entanglement spectra are calculated from infinite-W strips or cylinders ?

The infinite-W strip geometry was used. There, the low-lying contribution to the entanglement spectrum originates from the gapless top/bottom edge states as the bulk is gapped. We characterized these states by conformal field theory. The explicit geometry is now mentioned in Fig. 4.

Values of bond dimensions used are relatively small compared to state-of-the-art in VUMPS/iDMRG simulations of spin models. Is this due to the overhead in handling anyons ? What is the leading comp. complexity here - would, say, H=10 case be computationally too prohibitive ?

The complexity of the model comes from the plaquette term in Eq. (2) when mapped to 1D chain. Those terms, even though seem local in 2D, involve the braiding of $H+2$ legs. This increases the bond dimension of the MPO enormously leading to the limitation of both $\chi$ and $H$.

footnote 9(10 in the current manuscript) states " ... force $e_n=1,η$ for odd $n$ and $e_n=σ$ for even n". Isn't it the other way around, i.e. for $e_2$ the 1⊗σ⊗σ=1⊕η, so $e_n=1,η$ for even n ?

Thank you very much for pointing this out. We modified the footnote accordingly.

When discussing bulk phase, i.e. Fib⊠Spin(7)1 for θ/π=0.2 etc., these identifications are based on the entanglement spectra data (+ central charge) ? Or specifically, what (presumably emergent) structure in tensors A reveals this {1,η,σ} MTC ?

Our discussion is based on the entanglement spectra and central charge. In addition, the real-space chirality data in Fig. 1 is also considered: θ/π=0.2 and θ/π=0.9 have the opposite chirality and thus have the opposite sign in the central charge. (2.8-2 = 0.8 and 2.8-3.5 = -0.7)

Here, every site is taken to host a non-trivial object. Should the Ψ be seen as describing a number of WH anyons ρ, i.e. a "full filling" ? Can one introduce a chemical potential for ρ-anyons ?

It is an interesting thing to consider. In the current study, ρ is not a dynamical degree of freedom, so there is no direct correspondence to "full filling." For instance, the spinfull electrons no longer have dynamical degrees of freedom if its full filling while fusion tree Ψ can be still dynamical. However, we think the notion of filling might be introduced by considering ρ as a non-simple object e.g. ρ=1+σ. We expect that the physics changes accordingly, which would be interesting for the future studies.

1- Could the authors expand upon the ambiguity of the evaluation of hp? Is this due to various options in ordering of ρ edges ?

We have addressed the ambiguity in Footnote 5.

For θ/π=0.2 region in the inset of Fig. 3 (c) the fit for θ/π=0.1 slightly deviates from the data. Maybe adding θ/π = 0.3 and 0.4 for H=5 one can appreciate the distinction between situations where scaling formula (8) applies with high precision and cases where the edges remain gapped. [Here caption says inserts, main text insets]

Thank you very much for the suggestions. We have added the data points for θ/π = 0.3 and 0.4 and also slightly increased the max bond dimensions. It has become clear that those new data points are indeed distinct from the scaling regimes.

---

## Editorial Decision

accepted_in_target_journal